# Pointer Networks

**Oriol Vinyals**[*]
Google Brain

**Meire Fortunato**[*]
Department of Mathematics, UC Berkeley

**Navdeep Jaitly**
Google Brain

## Abstract

We introduce a new neural architecture to learn the conditional probability of an output sequence with elements that are discrete tokens corresponding to positions in an input sequence. Such problems cannot be trivially addressed by existent approaches such as sequence-to-sequence [1] and Neural Turing Machines [2], because the number of target classes in each step of the output depends on the length of the input, which is variable. Problems such as sorting variable sized sequences, and various combinatorial optimization problems belong to this class. Our model solves the problem of variable size output dictionaries using a recently proposed mechanism of neural attention. It differs from the previous attention attempts in that, instead of using attention to blend hidden units of an encoder to a context vector at each decoder step, it uses attention as a pointer to select a member of the input sequence as the output. We call this architecture a Pointer Net (Ptr-Net). We show Ptr-Nets can be used to learn approximate solutions to three challenging geometric problems – finding planar convex hulls, computing Delaunay triangulations, and the planar Travelling Salesman Problem – using training examples alone. Ptr-Nets not only improve over sequence-to-sequence with input attention, but also allow us to generalize to variable size output dictionaries. We show that the learnt models generalize beyond the maximum lengths they were trained on. We hope our results on these tasks will encourage a broader exploration of neural learning for discrete problems.

## 1   Introduction

Recurrent Neural Networks (RNNs) have been used for learning functions over sequences from examples for more than three decades [3]. However, their architecture limited them to settings where the inputs and outputs were available at a fixed frame rate (e.g. [4]). The recently introduced sequence-to-sequence paradigm [1] removed these constraints by using one RNN to map an input sequence to an embedding and another (possibly the same) RNN to map the embedding to an output sequence. Bahdanau et. al. augmented the decoder by propagating extra contextual information from the input using a content-based attentional mechanism [5, 2, 6, 7]. These developments have made it possible to apply RNNs to new domains, achieving state-of-the-art results in core problems in natural language processing such as translation [1, 5] and parsing [8], image and video captioning [9, 10], and even learning to execute small programs [2, 11].

Nonetheless, these methods still require the size of the output dictionary to be fixed *a priori*. Because of this constraint we cannot directly apply this framework to combinatorial problems where the size of the output dictionary depends on the length of the input sequence. In this paper, we address this limitation by repurposing the attention mechanism of [5] to create pointers to input elements. We show that the resulting architecture, which we name Pointer Networks (Ptr-Nets), can be trained to output satisfactory solutions to three combinatorial optimization problems – computing planar convex hulls, Delaunay triangulations and the symmetric planar Travelling Salesman Problem (TSP). The resulting models produce approximate solutions to these problems in a purely data driven fash-

---

[*]Equal contribution

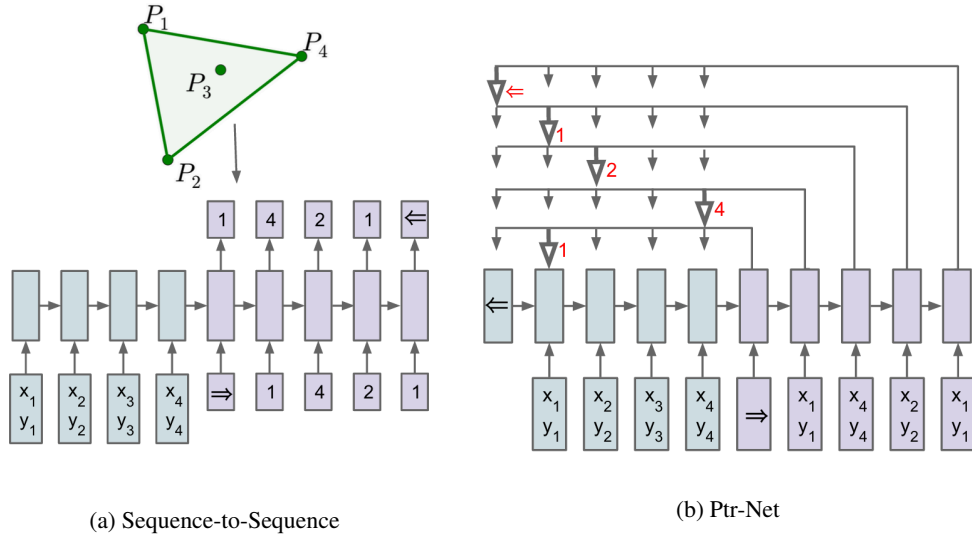

| (a) Sequence-to-Sequence | (b) Ptr-Net |

Figure 1: **(a)** *Sequence-to-Sequence* - An RNN (blue) processes the input sequence to create a code vector that is used to generate the output sequence (purple) using the probability chain rule and another RNN. The output dimensionality is fixed by the dimensionality of the problem and it is the same during training and inference [1]. **(b)** *Ptr-Net* - An encoding RNN converts the input sequence to a code (blue) that is fed to the generating network (purple). At each step, the generating network produces a vector that modulates a content-based attention mechanism over inputs ([5, 2]). The output of the attention mechanism is a softmax distribution with dictionary size equal to the length of the input.

ion (i.e., when we only have examples of inputs and desired outputs). The proposed approach is depicted in Figure 1.

The main contributions of our work are as follows:

- We propose a new architecture, that we call Pointer Net, which is simple and effective. It deals with the fundamental problem of representing variable length dictionaries by using a softmax probability distribution as a "pointer".
- We apply the Pointer Net model to three distinct non-trivial algorithmic problems involving geometry. We show that the learned model generalizes to test problems with more points than the training problems.
- Our Pointer Net model learns a competitive small scale ($n \leq 50$) TSP approximate solver. Our results demonstrate that a purely data driven approach can learn approximate solutions to problems that are computationally intractable.

## 2 Models

We review the sequence-to-sequence [1] and input-attention models [5] that are the baselines for this work in Sections 2.1 and 2.2. We then describe our model - Ptr-Net in Section 2.3.

### 2.1 Sequence-to-Sequence Model

Given a training pair, $(\mathcal{P}, \mathcal{C}^{\mathcal{P}})$, the sequence-to-sequence model computes the conditional probability $p(\mathcal{C}^{\mathcal{P}}|\mathcal{P}; \theta)$ using a parametric model (an RNN with parameters $\theta$) to estimate the terms of the probability chain rule (also see Figure 1), i.e.

$$p(\mathcal{C}^{\mathcal{P}}|\mathcal{P}; \theta) = \prod_{i=1}^{m(\mathcal{P})} p(C_i|C_1, \ldots, C_{i-1}, \mathcal{P}; \theta). \tag{1}$$

Here $\mathcal{P} = \{P_1, \ldots, P_n\}$ is a sequence of $n$ vectors and $\mathcal{C}^{\mathcal{P}} = \{C_1, \ldots, C_{m(\mathcal{P})}\}$ is a sequence of $m(\mathcal{P})$ indices, each between 1 and $n$ (we note that the target sequence length $m(\mathcal{P})$ is, in general, a function of $\mathcal{P}$).

The parameters of the model are learnt by maximizing the conditional probabilities for the training set, i.e.

$$\theta^* = \arg\max_{\theta} \sum_{\mathcal{P}, \mathcal{C}^{\mathcal{P}}} \log p(\mathcal{C}^{\mathcal{P}} | \mathcal{P}; \theta), \tag{2}$$

where the sum is over training examples.

As in [1], we use an Long Short Term Memory (LSTM) [12] to model $p(C_i | C_1, \ldots, C_{i-1}, \mathcal{P}; \theta)$. The RNN is fed $P_i$ at each time step, $i$, until the end of the input sequence is reached, at which time a special symbol, $\Rightarrow$ is input to the model. The model then switches to the generation mode until the network encounters the special symbol $\Leftarrow$, which represents termination of the output sequence.

Note that this model makes no statistical independence assumptions. We use two separate RNNs (one to encode the sequence of vectors $P_j$, and another one to produce or decode the output symbols $C_i$). We call the former RNN the encoder and the latter the decoder or the generative RNN.

During inference, given a sequence $\mathcal{P}$, the learnt parameters $\theta^*$ are used to select the sequence $\hat{\mathcal{C}}^{\mathcal{P}}$ with the highest probability, i.e., $\hat{\mathcal{C}}^{\mathcal{P}} = \arg\max_{\mathcal{C}^{\mathcal{P}}} p(\mathcal{C}^{\mathcal{P}} | \mathcal{P}; \theta^*)$. Finding the optimal sequence $\hat{\mathcal{C}}$ is computationally impractical because of the combinatorial number of possible output sequences. Instead we use a beam search procedure to find the best possible sequence given a beam size.

In this sequence-to-sequence model, the output dictionary size for all symbols $C_i$ is fixed and equal to $n$, since the outputs are chosen from the input. Thus, we need to train a separate model for each $n$. This prevents us from learning solutions to problems that have an output dictionary with a size that depends on the input sequence length.

Under the assumption that the number of outputs is $O(n)$ this model has computational complexity of $O(n)$. However, exact algorithms for the problems we are dealing with are more costly. For example, the convex hull problem has complexity $O(n \log n)$. The attention mechanism (see Section 2.2) adds more "computational capacity" to this model.

## 2.2 Content Based Input Attention

The vanilla sequence-to-sequence model produces the entire output sequence $\mathcal{C}^{\mathcal{P}}$ using the fixed dimensional state of the recognition RNN at the end of the input sequence $\mathcal{P}$. This constrains the amount of information and computation that can flow through to the generative model. The attention model of [5] ameliorates this problem by augmenting the encoder and decoder RNNs with an additional neural network that uses an attention mechanism over the entire sequence of encoder RNN states.

For notation purposes, let us define the encoder and decoder hidden states as $(e_1, \ldots, e_n)$ and $(d_1, \ldots, d_{m(\mathcal{P})})$, respectively. For the LSTM RNNs, we use the state after the output gate has been component-wise multiplied by the cell activations. We compute the attention vector at each output time $i$ as follows:

$$
\begin{aligned}
u_j^i &= v^T \tanh(W_1 e_j + W_2 d_i) \quad j \in (1, \ldots, n) \\
a_j^i &= \text{softmax}(u_j^i) \qquad\qquad\quad j \in (1, \ldots, n) \\
d_i' &= \sum_{j=1}^{n} a_j^i e_j
\end{aligned}
\tag{3}
$$

where softmax normalizes the vector $u^i$ (of length $n$) to be the "attention" mask over the inputs, and $v$, $W_1$, and $W_2$ are learnable parameters of the model. In all our experiments, we use the same hidden dimensionality at the encoder and decoder (typically 512), so $v$ is a vector and $W_1$ and $W_2$ are square matrices. Lastly, $d_i'$ and $d_i$ are concatenated and used as the hidden states from which we make predictions and which we feed to the next time step in the recurrent model.

Note that for each output we have to perform $n$ operations, so the computational complexity at inference time becomes $O(n^2)$.

This model performs significantly better than the sequence-to-sequence model on the convex hull problem, but it is not applicable to problems where the output dictionary size depends on the input.

Nevertheless, a very simple extension (or rather reduction) of the model allows us to do this easily.

### 2.3 Ptr-Net

We now describe a very simple modification of the attention model that allows us to apply the method to solve combinatorial optimization problems where the output dictionary size depends on the number of elements in the input sequence.

The sequence-to-sequence model of Section 2.1 uses a softmax distribution over a fixed sized output dictionary to compute $p(C_i|C_1, \ldots, C_{i-1}, \mathcal{P})$ in Equation 1. Thus it cannot be used for our problems where the size of the output dictionary is equal to the length of the input sequence. To solve this problem we model $p(C_i|C_1, \ldots, C_{i-1}, \mathcal{P})$ using the attention mechanism of Equation 3 as follows:

$$
\begin{aligned}
u_j^i &= v^T \tanh(W_1 e_j + W_2 d_i) \quad j \in (1, \ldots, n) \\
p(C_i|C_1, \ldots, C_{i-1}, \mathcal{P}) &= \mathrm{softmax}(u^i)
\end{aligned}
$$

where softmax normalizes the vector $u^i$ (of length $n$) to be an output distribution over the dictionary of inputs, and $v$, $W_1$, and $W_2$ are learnable parameters of the output model. Here, we do not blend the encoder state $e_j$ to propagate extra information to the decoder, but instead, use $u_j^i$ as pointers to the input elements. In a similar way, to condition on $C_{i-1}$ as in Equation 1, we simply copy the corresponding $P_{C_{i-1}}$ as the input. Both our method and the attention model can be seen as an application of content-based attention mechanisms proposed in [6, 5, 2, 7].

We also note that our approach specifically targets problems whose outputs are discrete and correspond to positions in the input. Such problems may be addressed artificially – for example we could learn to output the coordinates of the target point directly using an RNN. However, at inference, this solution does not respect the constraint that the outputs map back to the inputs exactly. Without the constraints, the predictions are bound to become blurry over longer sequences as shown in sequence-to-sequence models for videos [13].

## 3 Motivation and Datasets Structure

In the following sections, we review each of the three problems we considered, as well as our data generation protocol.[1]

In the training data, the inputs are planar point sets $\mathcal{P} = \{P_1, \ldots, P_n\}$ with $n$ elements each, where $P_j = (x_j, y_j)$ are the cartesian coordinates of the points over which we find the convex hull, the Delaunay triangulation or the solution to the corresponding Travelling Salesman Problem. In all cases, we sample from a uniform distribution in $[0, 1] \times [0, 1]$. The outputs $\mathcal{C}^{\mathcal{P}} = \{C_1, \ldots, C_{m(\mathcal{P})}\}$ are sequences representing the solution associated to the point set $\mathcal{P}$. In Figure 2, we find an illustration of an input/output pair $(\mathcal{P}, \mathcal{C}^{\mathcal{P}})$ for the convex hull and the Delaunay problems.

### 3.1 Convex Hull

We used this example as a baseline to develop our models and to understand the difficulty of solving combinatorial problems with data driven approaches. Finding the convex hull of a finite number of points is a well understood task in computational geometry, and there are several exact solutions available (see [14, 15, 16]). In general, finding the (generally unique) solution has complexity $O(n \log n)$, where $n$ is the number of points considered.

The vectors $\mathcal{P}_j$ are uniformly sampled from $[0, 1] \times [0, 1]$. The elements $C_i$ are indices between 1 and $n$ corresponding to positions in the sequence $\mathcal{P}$, or special tokens representing beginning or end of sequence. See Figure 2 (a) for an illustration. To represent the output as a sequence, we start from the point with the lowest index, and go counter-clockwise – this is an arbitrary choice but helps reducing ambiguities during training.

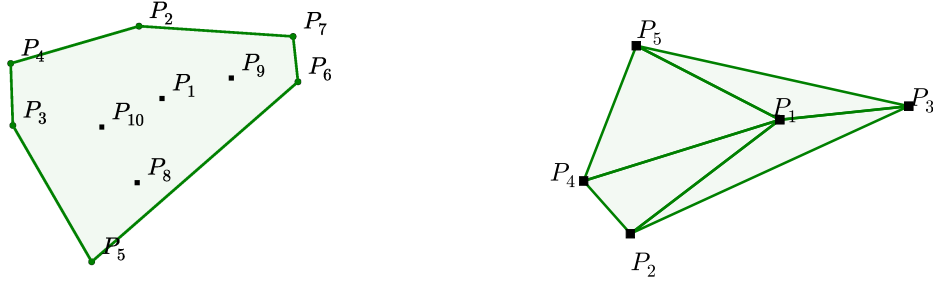

(a) Input $\mathcal{P} = \{P_1, \ldots, P_{10}\}$, and the output sequence $\mathcal{C}^{\mathcal{P}} = \{\Rightarrow, 2, 4, 3, 5, 6, 7, 2, \Leftarrow\}$ representing its convex hull.

(b) Input $\mathcal{P} = \{P_1, \ldots, P_5\}$, and the output $\mathcal{C}^{\mathcal{P}} = \{\Rightarrow, (1, 2, 4), (1, 4, 5), (1, 3, 5), (1, 2, 3), \Leftarrow\}$ representing its Delaunay Triangulation.

Figure 2: Input/output representation for (**a**) convex hull and (**b**) Delaunay triangulation. The tokens $\Rightarrow$ and $\Leftarrow$ represent beginning and end of sequence, respectively.

## 3.2 Delaunay Triangulation

A Delaunay triangulation for a set $\mathcal{P}$ of points in a plane is a triangulation such that each circumcircle of every triangle is empty, that is, there is no point from $\mathcal{P}$ in its interior. Exact $O(n \log n)$ solutions are available [17], where $n$ is the number of points in $\mathcal{P}$.

In this example, the outputs $\mathcal{C}^{\mathcal{P}} = \{C_1, \ldots, C_{m(\mathcal{P})}\}$ are the corresponding sequences representing the triangulation of the point set $\mathcal{P}$. Each $C_i$ is a triple of integers from 1 to $n$ corresponding to the position of triangle vertices in $\mathcal{P}$ or the beginning/end of sequence tokens. See Figure 2 (b).

We note that any permutation of the sequence $\mathcal{C}^{\mathcal{P}}$ represents the same triangulation for $\mathcal{P}$, additionally each triangle representation $C_i$ of three integers can also be permuted. Without loss of generality, and similarly to what we did for convex hulls at training time, we order the triangles $C_i$ by their incenter coordinates (lexicographic order) and choose the increasing triangle representation[2]. Without ordering, the models learned were not as good, and finding a better ordering that the Ptr-Net could better exploit is part of future work.

## 3.3 Travelling Salesman Problem (TSP)

TSP arises in many areas of theoretical computer science and is an important algorithm used for microchip design or DNA sequencing. In our work we focused on the planar symmetric TSP: given a list of cities, we wish to find the shortest possible route that visits each city exactly once and returns to the starting point. Additionally, we assume the distance between two cities is the same in each opposite direction. This is an NP-hard problem which allows us to test the capabilities and limitations of our model.

The input/output pairs $(\mathcal{P}, \mathcal{C}^{\mathcal{P}})$ have a similar format as in the Convex Hull problem described in Section 3.1. $\mathcal{P}$ will be the cartesian coordinates representing the cities, which are chosen randomly in the $[0, 1] \times [0, 1]$ square. $\mathcal{C}^{\mathcal{P}} = \{C_1, \ldots, C_n\}$ will be a permutation of integers from 1 to $n$ representing the optimal path (or tour). For consistency, in the training dataset, we always start in the first city without loss of generality.

To generate exact data, we implemented the Held-Karp algorithm [18] which finds the optimal solution in $O(2^n n^2)$ (we used it up to $n = 20$). For larger $n$, producing exact solutions is extremely costly, therefore we also considered algorithms that produce approximated solutions: A1 [19] and A2 [20], which are both $O(n^2)$, and A3 [21] which implements the $O(n^3)$ Christofides algorithm. The latter algorithm is guaranteed to find a solution within a factor of 1.5 from the optimal length. Table 2 shows how they performed in our test sets.

# 4 Empirical Results

## 4.1 Architecture and Hyperparameters

No extensive architecture or hyperparameter search of the Ptr-Net was done in the work presented here, and we used virtually the same architecture throughout all the experiments and datasets. Even though there are likely some gains to be obtained by tuning the model, we felt that having the same model hyperparameters operate on all the problems makes the main message of the paper stronger.

As a result, all our models used a single layer LSTM with either 256 or 512 hidden units, trained with stochastic gradient descent with a learning rate of 1.0, batch size of 128, random uniform weight initialization from -0.08 to 0.08, and L2 gradient clipping of 2.0. We generated 1M training example pairs, and we did observe overfitting in some cases where the task was simpler (i.e., for small $n$). Training generally converged after 10 to 20 epochs.

## 4.2 Convex Hull

We used the convex hull as the guiding task which allowed us to understand the deficiencies of standard models such as the sequence-to-sequence approach, and also setting up our expectations on what a purely data driven model would be able to achieve with respect to an exact solution.

We reported two metrics: accuracy, and area covered of the true convex hull (note that any simple polygon will have full intersection with the true convex hull). To compute the accuracy, we considered two output sequences $\mathcal{C}^1$ and $\mathcal{C}^2$ to be the same if they represent the same polygon. For simplicity, we only computed the area coverage for the test examples in which the output represents a simple polygon (i.e., without self-intersections). If an algorithm fails to produce a simple polygon in more than 1% of the cases, we simply reported FAIL.

The results are presented in Table 1. We note that the area coverage achieved with the Ptr-Net is close to 100%. Looking at examples of mistakes, we see that most problems come from points that are aligned (see Figure 3 (d) for a mistake for $n = 500$) – this is a common source of errors in most algorithms to solve the convex hull.

It was seen that the order in which the inputs are presented to the encoder during inference affects its performance. When the points on the true convex hull are seen "late" in the input sequence, the accuracy is lower. This is possibly the network does not have enough processing steps to "update" the convex hull it computed until the latest points were seen. In order to overcome this problem, we used the attention mechanism described in Section 2.2, which allows the decoder to look at the whole input at any time. This modification boosted the model performance significantly. We inspected what attention was focusing on, and we observed that it was "pointing" at the correct answer on the input side. This inspired us to create the Ptr-Net model described in Section 2.3.

More than outperforming both the LSTM and the LSTM with attention, our model has the key advantage of being inherently variable length. The bottom half of Table 1 shows that, when training our model on a variety of lengths ranging from 5 to 50 (uniformly sampled, as we found other forms of curriculum learning to not be effective), a single model is able to perform quite well on all lengths it has been trained on (but some degradation for $n = 50$ can be observed w.r.t. the model trained only on length 50 instances). More impressive is the fact that the model does extrapolate to lengths that it has never seen during training. Even for $n = 500$, our results are satisfactory and indirectly indicate that the model has learned more than a simple lookup. Neither LSTM or LSTM with attention can be used for any given $n' \neq n$ without training a new model on $n'$.

## 4.3 Delaunay Triangulation

The Delaunay Triangulation test case is connected to our first problem of finding the convex hull. In fact, the Delaunay Triangulation for a given set of points triangulates the convex hull of these points.

We reported two metrics: accuracy and triangle coverage in percentage (the percentage of triangles the model predicted correctly). Note that, in this case, for an input point set $\mathcal{P}$, the output sequence $\mathcal{C}(\mathcal{P})$ is, in fact, a set. As a consequence, any permutation of its elements will represent the same triangulation.

Table 1: Comparison between LSTM, LSTM with attention, and our Ptr-Net model on the convex hull problem. Note that the baselines must be trained on the same $n$ that they are tested on. 5-50 means the dataset had a uniform distribution over lengths from 5 to 50.

| METHOD | TRAINED $n$ | $n$ | ACCURACY | AREA |
|---|---|---|---|---|
| LSTM [1] | 50 | 50 | 1.9% | FAIL |
| +ATTENTION [5] | 50 | 50 | 38.9% | 99.7% |
| PTR-NET | 50 | 50 | 72.6% | 99.9% |
| LSTM [1] | 5 | 5 | 87.7% | 99.6% |
| PTR-NET | 5-50 | 5 | 92.0% | 99.6% |
| LSTM [1] | 10 | 10 | 29.9% | FAIL |
| PTR-NET | 5-50 | 10 | 87.0% | 99.8% |
| PTR-NET | 5-50 | 50 | 69.6% | 99.9% |
| PTR-NET | 5-50 | 100 | 50.3% | 99.9% |
| PTR-NET | 5-50 | 200 | 22.1% | 99.9% |
| PTR-NET | 5-50 | 500 | 1.3% | 99.2% |

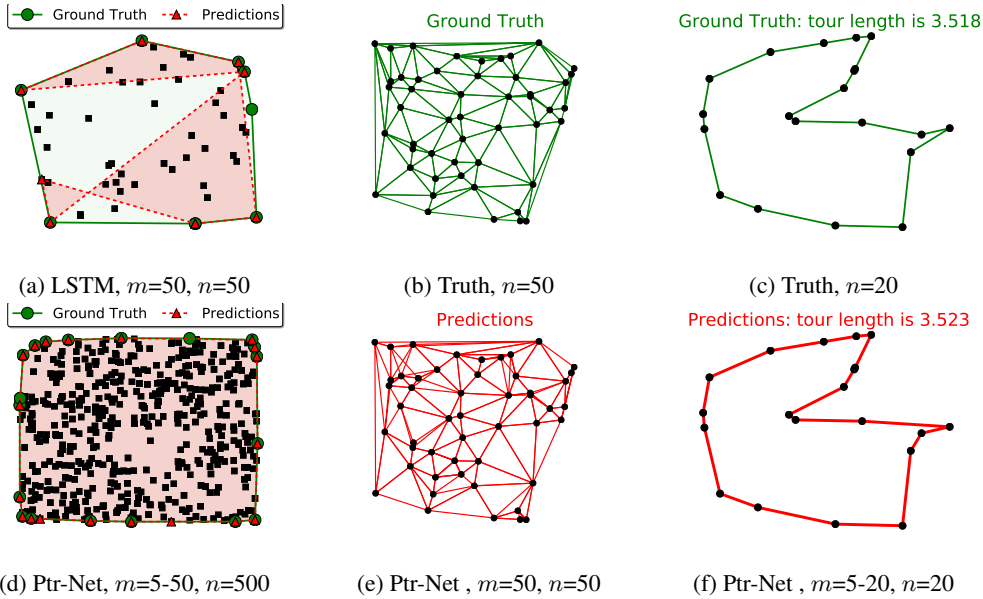

(a) LSTM, $m$=50, $n$=50  (b) Truth, $n$=50  (c) Truth, $n$=20

(d) Ptr-Net, $m$=5-50, $n$=500  (e) Ptr-Net , $m$=50, $n$=50  (f) Ptr-Net , $m$=5-20, $n$=20

Figure 3: Examples of our model on Convex hulls (left), Delaunay (center) and TSP (right), trained on $m$ points, and tested on $n$ points. A failure of the LSTM sequence-to-sequence model for Convex hulls is shown in (a). Note that the baselines cannot be applied to a different length from training.

Using the Ptr-Net model for $n = 5$, we obtained an accuracy of 80.7% and triangle coverage of 93.0%. For $n = 10$, the accuracy was 22.6% and the triangle coverage 81.3%. For $n = 50$, we did not produce any precisely correct triangulation, but obtained 52.8% triangle coverage. See the middle column of Figure 3 for an example for $n = 50$.

## 4.4 Travelling Salesman Problem

We considered the planar symmetric travelling salesman problem (TSP), which is NP-hard as the third problem. Similarly to finding convex hulls, it also has sequential outputs. Given that the Ptr-Net implements an $O(n^2)$ algorithm, it was unclear if it would have enough capacity to learn a useful algorithm solely from data.

As discussed in Section 3.3, it is feasible to generate exact solutions for relatively small values of $n$ to be used as training data. For larger $n$, due to the importance of TSP, good and efficient algorithms providing reasonable approximate solutions exist. We used three different algorithms in our experiments – A1, A2, and A3 (see Section 3.3 for references).

Table 2: Tour length of the Ptr-Net and a collection of algorithms on a small scale TSP problem.

| $n$ | OPTIMAL | A1 | A2 | A3 | PTR-NET |
|---|---|---|---|---|---|
| 5 | 2.12 | 2.18 | 2.12 | 2.12 | 2.12 |
| 10 | 2.87 | 3.07 | 2.87 | 2.87 | 2.88 |
| 50 (A1 TRAINED) | N/A | 6.46 | 5.84 | 5.79 | 6.42 |
| 50 (A3 TRAINED) | N/A | 6.46 | 5.84 | 5.79 | 6.09 |
| 5 (5-20 TRAINED) | 2.12 | 2.18 | 2.12 | 2.12 | 2.12 |
| 10 (5-20 TRAINED) | 2.87 | 3.07 | 2.87 | 2.87 | 2.87 |
| 20 (5-20 TRAINED) | 3.83 | 4.24 | 3.86 | 3.85 | 3.88 |
| 25 (5-20 TRAINED) | N/A | 4.71 | 4.27 | 4.24 | 4.30 |
| 30 (5-20 TRAINED) | N/A | 5.11 | 4.63 | 4.60 | 4.72 |
| 40 (5-20 TRAINED) | N/A | 5.82 | 5.27 | 5.23 | 5.91 |
| 50 (5-20 TRAINED) | N/A | 6.46 | 5.84 | 5.79 | 7.66 |

Table 2 shows all of our results on TSP. The number reported is the length of the proposed tour. Unlike the convex hull and Delaunay triangulation cases, where the decoder was unconstrained, in this example we set the beam search procedure to only consider valid tours. Otherwise, the Ptr-Net model would sometimes output an invalid tour – for instance, it would repeat two cities or decided to ignore a destination. This procedure was relevant for $n > 20$: for $n \leq 20$, the unconstrained decoding failed less than 1% of the cases, and thus was not necessary. For 30, which goes beyond the longest sequence seen in training, failure rate went up to 35%, and for 40, it went up to 98%.

The first group of rows in the table show the Ptr-Net trained on optimal data, except for $n = 50$, since that is not feasible computationally (we trained a separate model for each $n$). Interestingly, when using the worst algorithm (A1) data to train the Ptr-Net, our model outperforms the algorithm that is trying to imitate.

The second group of rows in the table show how the Ptr-Net trained on optimal data with 5 to 20 cities can generalize beyond that. The results are virtually perfect for $n = 25$, and good for $n = 30$, but it seems to break for 40 and beyond (still, the results are far better than chance). This contrasts with the convex hull case, where we were able to generalize by a factor of 10. However, the underlying algorithms have greater complexity than $O(n \log n)$, which could explain this.

## 5 Conclusions

In this paper we described Ptr-Net, a new architecture that allows us to learn a conditional probability of one sequence $\mathcal{C}^{\mathcal{P}}$ given another sequence $\mathcal{P}$, where $\mathcal{C}^{\mathcal{P}}$ is a sequence of discrete tokens corresponding to positions in $\mathcal{P}$. We show that Ptr-Nets can be used to learn solutions to three different combinatorial optimization problems. Our method works on variable sized inputs (yielding variable sized output dictionaries), something the baseline models (sequence-to-sequence with or without attention) cannot do directly. Even more impressively, they outperform the baselines on fixed input size problems - to which both the models can be applied.

Previous methods such as RNNSearch, Memory Networks and Neural Turing Machines [5, 6?  ] have used attention mechanisms to process inputs. However these methods do not directly address problems that arise with variable output dictionaries. We have shown that an attention mechanism can be applied to the output to solve such problems. In so doing, we have opened up a new class of problems to which neural networks can be applied without artificial assumptions. In this paper, we have applied this extension to RNNSearch, but the methods are equally applicable to Memory Networks and Neural Turing Machines.

Future work will try and show its applicability to other problems such as sorting where the outputs are chosen from the inputs. We are also excited about the possibility of using this approach to other combinatorial optimization problems.

### Acknowledgments

We would like to thank Rafal Jozefowicz, Ilya Sutskever, Quoc Le and Samy Bengio for useful discussions. We would also like to thank Daniel Gillick for his help with the final manuscript.

## Footnotes

[1] We will release all the datasets at `hidden` for reference.

[2]We choose $C_i = (1, 2, 4)$ instead of (2,4,1) or any other permutation.

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
