[Reviews · NeurIPS 2015]

Submitted by Assigned_Reviewer_1

Authors of this paper propose an attention model based on [1] by using LSTMs. Their model decides what to predict by using the output of a softmax MLP over the inputs as opposed to regular attention model used in [1](RNNSearch), where they used that to use for the convex combination of the input annotations to predict where to attend to translate in the target language. The output of the softmax-MLP predicts the location of the output at each timestep directly from the input sequence. The advantage of this approach is that, the output softmax dictionary only depends on the length of the input sequence instead of on the number of possible instances in the problem. The authors have provided very interesting experimental results on three different discrete optimization problems.

1) In the introduction authors should cite [2] for attention with image caption generation. 2) For attention mechanism you should probably cite [3] as well. 3) On Page 3, m(P) is being used without any clear explanation of it in the text. 4) On Page 3, line 130. In sequence to sequence model for each n, you do not really have to train a different model. You could just train a different softmax for each n and the rest of the parameters can be shared for all the problems of different lengths. 5) Why you didn't use bidirectional RNN on the inputs for both "Content Based Input Attention" and "Ptr-Net". 6) Have you tried using deep LSTMs for encoder and decoder and observe its effect/importance for problems of different difficulties? 7) In section 2.3, you should add equation number there as well. Such that in line 183 you could change Equation to "Equation 2" instead.

8) For Delaunay Triangulation problem ordering seems to be important. Do you have any justification on why lexicographic ordering performed better? 9) What have you done with the input dictionary size? Did you also restrict it? Did you use one-hot representation for the input? Probably some of these problems, you do not even need to learn embeddings for the input, you might very well use binary vector representation instead. 10) On line 238, sentence starting with "Exact O(nlogn) solution...", does that mean that there are O(nlogn) unique number of triangulations for a particular graph?

11) Does each set of points in your training set for Delanuay Triangulation problem has only a single solution? 12) The sentence starting on line 307 is confusing. LSTMs and RNNs can also used with variable length sequences as well. I think, you try to illustrate that the output softmax output size of your model depends only on the length of the sentence. That should be clarified. 13) On Table 1, the notation "5-50" is never explained in the text. Does that mean that you train both of on sequences of length 5 and 50? 14) Can you provide a plot of performance of your model with respect to the length of the training sequences? 15) For TSP, you had to restrict your beam-search to only valid sequences. This means that the pointer network was not able to capture the concept of TSP. Possibly this kind of behavior with the current architecture do not automatically pop-out with gradient descent training and using backprop. Have you consider an architectural prior or some form of constraint(may be a penalty on your softmax output) on your model to prevent it to generate non-valid sequences? On Table 2, I think you should at least add the results without constraining the beam-search to valid sequences.

16) Have you considered doing convex-hull in higher dimensions?

A minor comment, I am not completely confident whether if the authors have enough novelty with their architecture over RNNSearch to name their model as Pointer Networks.

If authors accept to address 15 [putting results without limiting beam-search], I am willing to increase my score 7 or 8. In the end, this is a very interesting and important topic, which needs to be studied more extensively by machine learning community.

[1] Bahdanau, Dzmitry, Kyunghyun Cho, and Yoshua Bengio. "Neural machine translation by jointly learning to align and translate." arXiv preprint arXiv:1409.0473 (2014). [2] Xu, Kelvin, et al. "Show, attend and tell: Neural image caption generation with visual attention." arXiv preprint arXiv:1502.03044 (2015). [3] Graves, Alex. "Generating sequences with recurrent neural networks." arXiv preprint arXiv:1308.0850 (2013).
Summary: This is a well written paper with very interesting experimental results.

Submitted by Assigned_Reviewer_2

This paper addresses the question of using neural networks to solve computational geometry problems, such as Traveling Salesman Problem (TSP), finding the convex hull in the plane, or finding Delaunay triangulations. To that end, the authors propose a slight variant of recent content-based attention mechanisms allowing them to predict a sequence of positions (or indices) of the input sequence, and thus train on the aforementioned tasks. Numerical results show that their model learns approximate solutions and is able to generalize beyond the sizes used during training.

The main contribution of this paper is the problem set-up, casting some fundamental computational geometry problems as particular instances of "sequence-to-sequence" tasks, similarly as other recent efforts (such as "Learning to Execute" by Zaremba & Sutskever), and asking whether data-driven algorithms can compete with their analytical counterparts. Besides the practical interest (which might be relevant for the TSP, since it is NP hard and therefore we typically rely on approximate algorithms), this is a scientifically interesting question.

This work might serve as motivation and benchmark for future efforts in that direction.

The paper is clearly written (although some aspects are discussed in too broad terms cf below comments), the set-up is well presented and the three problems are described with enough detail. As I mentioned before, the paper is original in its set-up, less so in its proposed architecture. As pointed out by the authors themselves, the so-called Pointer Networks are a minor variant of the attention model of [5], so that it can be used in the setting where the output of the network are the attention positions themselves rather than some exterior targets.

However, despite these positive aspects, I would like to point out some comments, in the hope to see them addressed in the rebuttal:

--> Although it is true that some recent architectures are unable to adapt to variable output dictionary size, my impression is that other works do permit this setup, such as "Inferring Algorithmic Patterns with stack-augmented recurrent nets", by A.Joulin and T.Mikolov, or the Memory networks by J.Weston. If that is the case, it would be interesting to compare the performance of these methods and to report the results.

--> Training procedure. A convex hull is presented here as a sequence of input points. However, there is a natural equivalence class in the output space, since the starting point and the direction are arbitrary. After the first two points of the output are known, the conditional probabilities of the next point do not suffer from this ambiguity, but couldn't the model be improved by resolving the ambiguity? In Section 3.2, where the outputs define an even larger equivalence class, the authors choose a lexicographic ordering (thus choosing one representative of the equivalence class). Why not do the same thing for the convex hull?

--> I found the analysis of the results to be a bit superficial, I would have liked to see more introspection into what is happening. For example, in Section 4.2, the authors stress the fact that their model generalizes to larger sequence lengths. My impression is that, while true, this claim should be studied more carefully. In this task, there are two essential cardinal quantities, the length n of the input sequence, and the size k of the convex hull. In the setup of points distributed iid in the square, k is typically of the order of o(4 log n) (see for example "On the expected complexity of random convex hulls" ), and it is unclear which quantity is more relevant to measure the generalization of the model, n or k. The latter typically grows at logarithmic rate. --> Lines 403-404 mentions one of the most interesting numerical results of the paper, namely that the network seems to "regularize" the A1 algorithm. Why? Even if it is a conjecture, this seems to be one of the essential messages of the paper, replacing a priori operations by data-driven modules.

--> Lastly, in lines 406-410, the authors compare the generalization performance of the network between the convex hull problem and the TSP. They explain the worse generalization of the TSP case from the complexity of the underlying algorithm (NP-complete vs n log(n)). While this is a valid point, I am wondering if there is another even more important aspect, namely the effective size of the output. In the case of the convex hull, as mentioned earlier, the typical size of the set is o(4 log n), and thus the number of bits to encode the output solution is of the order of log ( binomial (n, 4log n) ) = o( (log n)^2 ), whereas for the TSP, the output is essentially any permutation of the input, which requires of the order of log (n !) = n log(n) bits. Therefore, one could argue that in the convex hull case, the network needs to learn over a space of outputs which is of significantly smaller dimension than in the TSP. The fact is that there exsit o(n^2) approximate algorithms (A2) that do well -- but learning those operations over a large output space is challenging.

--> A fundamental tool to fight curse of dimensionality and NP-hardness is to rely on dynamic programming or multiscale tools. The paper proposes a very small variant of a readily available model, which has the advantage of being simple, but does not address this property. However, I am wondering if the interest of this line of work is precisely the class of intractable problems, in which being able to learn this multiscale structure seems a necessary step (similar in spirit to the fast multipole transform).

--> Even if it is not in the spirit of what is presented here, a natural question is why not try to learn the TSP by optimizing the length of the path (assuming one could backpropagate a continuous relaxation), or perhaps a fine-tuning step similar to a Viterbi decoding.

Summary: This paper discusses how to use neural networks to approach combinatorial optimization problems arising in computational geometry, such as Traveling Salesman or Convex hull. For that purpose, the authors propose a small modification of recent content-based attention mechanisms, enabling to learn subsequences of a discrete input stream.

The paper has the merit to show that neural networks, with very little assumptions of the problem at hand, can nearly match handcrafted algorithms at those tasks. However, the experimental section of the paper is quite weak, without comparisons to recent baselines (such as memory networks). Also, I found the analysis of results rather half-baked. Overall, I think the paper has still much room for improvement (hopefully done thanks to the rebuttal), and its contribution is valuable to the community, therefore I rate it just above the acceptance threshold.

Submitted by Assigned_Reviewer_3

The paper introduces a new type of recurrent neural network extending the attention based sequence-to-sequence model. It is able to handle output sequences with discrete tokens corresponding to positions in the input sequence, thus a variable length dictionary.

The idea is a simple modification of the input attention models, but opens the possibility to tackle new classes of tasks (in the paper geometric problems) which I find really interesting.

The paper is very well written and easy to follow.

Experiments are conducted on three challenging problems and

evaluation is sufficient in my point of view. In table 1 though I would report LSTM results training the model

for the longest sequence; i.e. I see PTR-NET 5-50 but only LSTM 5 and 10, why not 50?

Data used for the experiments will be released upon publication. Even if it will be not hard to produce such data I believe that having

it will definitely facilitate future research and comparisons. I suggest the evaluation software to be released as well as this is really important for reproducibility of research.
Summary: This paper is a simple extension of attention based sequence to sequence model; yet it opens up the possibility of tackling new classes of problems such as geometric problems.

It is well written, easy to follow and experiments are nicely done.

Submitted by Assigned_Reviewer_4

Personally, I find the idea of attacking combinatorial optimization using recurrent neural networks (RNNs) very interesting. I'm not aware of prior work in this specific direction, so I judge the originality of the work to be high. However, as it stands, the paper starts out by saying that we'd like to extend RNNs to such problems, then proposes a model and evaluates it. It's not made clear what the objective/s behind this exercise is/are. A popular mindset is to use learning methods for problems for which we can't explicitly program computers, so why is it interesting to study learning methods for problems which have known solution algorithms? The study should be put in perspective by citing earlier work using RNNs to learn algorithms for various tasks, and discussing the importance of understanding the power of these models. Some older work is on RNNs learning to run entire

learning algorithms e.g. [1,2] and there might be other similar motivating studies.

The introduction states that the motivation behind the proposed model is to extend the sequence-to-sequence architectures to combinatorial problems with varying sizes. Is it not possible to solve such problems using other RNN architectures (at least in principle)? It should be clarified that in principle a large RNN could be trained using a global optimization method to produce the desired solutions, producing the indices one-by-one at the same unit, or at multiple units using a binary encoding. The proposed model, however, can be trained with gradient descent to its (likely large) advantage.

In general I found the paper to be clear and well written. While I feel that this contribution is significant, I do feel strongly that the paper does not do a good job of discussing its own significance and putting it in context.

[1] Hochreiter, S., Younger, A. S., & Conwell, P. R. (2001). Learning to learn using gradient descent. In Artificial Neural Networks-ICANN 2001 (pp. 87-94). Springer Berlin Heidelberg. [2] J.

Schmidhuber. A self-referential weight matrix. In Proceedings of the International Conference on Artificial Neural Networks, Amsterdam, pages 446-451. Springer, 1993

The LSTM apparently uses forget gates which were not yet present in reference [11] but introduced by Gers et al (2000).

Specific Comments:

The experimental results are interesting and presented well. I would like to see more details about the training/validation/test sets. In particular, the sizes of the datasets used and the amount of time/epochs taken to train on these problems, perhaps with training curves. This information would be useful for readers and those attempting to reproduce the results.

It is interesting to see that for n=500 on the convex hull problem, the accuracy is very low but the area coverage is very high. If it makes mistakes similar to common algorithms for this task (as stated by the authors), it has perhaps implemented a similar algorithm? There is no reason why this must happen, and indeed for the TSP, the model sometimes performs better than the approximate algorithm used to train it. Together these results may indicate something deeper about the nature of algorithms these models learn, or prefer to learn.

Were problems in more than 2 dimensions attempted?
Summary: This paper introduces a variant of the encoder-decoder recurrent network architectures. The main motivation is to address problems for which the outputs are discrete symbols from a dictionary whose size is equal to the length of the input, instead of having a fixed size. The proposed model retains the additional computational complexity offered by an attention mechanism, but foregoes the mapping to a fixed size output dictionary. Instead, the attention "mask" over the input sequence is directly interpreted as the output. The model is applied to 3 combinatorial optimization problems with interesting results.

Author Feedback
Author rebuttal: We thank the reviewers for their comments.
We will add related work pointed by R1 and R3, and release data and evaluation code (R3, R5).

R1:
1,2,3,4,7# Agreed.
5# On using bidirectional RNN. Since we condition on the full sequence by conditioning on the last hidden state of the forward RNN, we do not believe BRNN will help.
6# RE architecture search. Other than choosing larger models for complex problems we did not experiment with architectures. We want to emphasize that not much tuning was needed after we implemented our modification to RNNSearch. Better results could be achieved with more tuning.
8# On ordering in Delaunay. Ordering also helped with convex hulls (starting at the lowest index, we go counterclockwise). This helps resolve the ambiguities in the answers and makes learning easier by keeping more correlated targets closer.
9# On input dictionary. We use the 2D coordinates of the selected point in the previous timestep instead of a discrete representation. We did not explore other representations (e.g. binary).
10,11# Delaunay Triangulation is unique (except if points exactly lie on a circle). O(nlogn) in line 238 refers to the running cost of the algorithm.
12# Yes, we should emphasize that our proposed architecture can deal with variable length softmax.
13# 5-50 notation means we train on a uniform distribution of lengths from 5 to 50.
14# We have such plots, and we'll try to squeeze it in the final version. The tables (for e.g. TSP) show this behavior well.
15# On restricting the decoder to valid tours for TSP. This is, indeed, an interesting and intriguing property of the model. We will add the numbers without the constrained search to the table. When not using constraints, errors typically occur when extrapolating to lengths beyond what the model was trained on. For the 5-20 TSP, if tested on 20 the failure rate is less than 1%, whereas for 30 it is 35%, and for 40 it is 98% (failure rate is less than 1% when constraining the search). These mistakes typically involve repeating a city twice instead of outputting two cities that are closeby. Indeed, some fixes could be made to inhibit return during training.
16# Yes. Finding Delaunay triangulation in D dimensions can be casted as finding convex-hulls in D+1 dimensions. It would be interesting to see if the model would also be able to perform this conversion.

R2:
1# RE comparison with Stack RNN, Memory nets. As admitted by the authors of Memory Nets, their architecture is essentially the same as RNNSearch. These models also use content based attention and RNNs. Our modification allows to "point" rather than retrieve some memory (thus the name), but we did compare against vanilla seq2seq and RNNSearch, for which we have state-of-the-art implementations on other tasks. We feel that the Stack RNN would probably suffer the same issues than RNNSearch - we could push and pop indices, but we would not be able to go beyond indices that have been seen in training. A discussion on the similarities will be added.
2# On the equivalence class of outputs. We chose this ordering to resolve the ambiguity for convex hull. See #8 from R1.
3,5# On lack of analysis. Due to space limitations, we removed some test cases that would partially answer your question. We will add these to the appendix and address the 'n vs k' issue. For instance, some mistakes are related to points that are aligned (similar to standard algorithms), regardless of k.
4# On improving A1 algorithm. Indeed we hypothesized that the RNN prior somehow helps produce better tours. We'll look for examples where there is a revealing improvement.
6# On multiscale approaches and NP-hard problems. We argue that attention is multiscale (as it can reprocess the input multiple times).
7# Directly optimizing length in TSP. We are investigating this on a problem that is simple to relaxate (3SAT), and directly optimizing the metric of interest is possible (in general, it is hard to find a smooth proxy of the cost).

R3:
On motivating of our work # Our main motivation and what we are able to show is that RNNs can approximate solutions to complicated problems in a data driven fashion. For academic purposes, we did use problems for which there are known algorithms that are well studied. But our goal is for the RNN to learn to solve problems for which we do NOT have a good algorithm, indeed. In this instance, producing answers may be costly (but possible, thanks to e.g. parallelization or perturbation of known solutions), and one could produce a training set of enough size from which an RNN could then learn from. We can emphasize this further in the introduction / conclusions.
RE algorithm learned # It is possible that it generates a similar algorithm by capturing the statistics of the solutions. Also see R2, #3-5.
More than 2D# We didn't, see R1, #16.

R456:
Table 1# LSTM 50 is the first row.
Memory Nets comparison # See R2, #1.